# In Vitro Release of Bioactive Bone Morphogenetic Proteins (GDF5, BB-1, and BMP-2) from a PLGA Fiber-Reinforced, Brushite-Forming Calcium Phosphate Cement

**DOI:** 10.3390/pharmaceutics11090455

**Published:** 2019-09-03

**Authors:** Francesca Gunnella, Elke Kunisch, Victoria Horbert, Stefan Maenz, Jörg Bossert, Klaus D. Jandt, Frank Plöger, Raimund W. Kinne

**Affiliations:** 1Experimental Rheumatology Unit, Department of Orthopedics, Jena University Hospital, Waldkrankenhaus “Rudolf Elle”, Klosterlausnitzer Str. 81, 07607 Eisenberg, Germany; 2Chair of Materials Science, Otto Schott Institute of Materials Research, Friedrich Schiller University Jena, 07743 Jena, Germany; 3Jena Center for Soft Matter (JCSM), Friedrich Schiller University Jena, 07743 Jena, Germany; 4Jena School for Microbial Communication (JSMC), Friedrich Schiller University Jena, 07743 Jena, Germany; 5BIOPHARM GmbH, 69115 Heidelberg, Germany

**Keywords:** in vitro release, bioactive bone morphogenetic proteins, GDF5, BB-1, BMP-2, PLGA fiber-reinforced, brushite-forming calcium phosphate cement

## Abstract

Bone regeneration of sheep lumbar osteopenia is promoted by targeted delivery of bone morphogenetic proteins (BMPs) via a biodegradable, brushite-forming calcium-phosphate-cement (CPC) with stabilizing poly(l-lactide-*co*-glycolide) acid (PLGA) fibers. The present study sought to quantify the release and bioactivity of BMPs from a specific own CPC formulation successfully used in previous in vivo studies. CPC solid bodies with PLGA fibers (0%, 5%, 10%) containing increasing dosages of GDF5, BB-1, and BMP-2 (2 to 1000 µg/mL) were ground and extracted in phosphate-buffered saline (PBS) or pure sheep serum/cell culture medium containing 10% fetal calf serum (FCS; up to 30/31 days). Released BMPs were quantified by ELISA, bioactivity was determined via alkaline phosphatase (ALP) activity after 3-day exposure of different osteogenic cell lines (C2C12; C2C12BRlb with overexpressed BMP-receptor-1b; MCHT-1/26; ATDC-5) and via the influence of the extracts on the expression of osteogenic/chondrogenic genes and proteins in human adipose tissue-derived mesenchymal stem cells (hASCs). There was hardly any BMP release in PBS, whereas in medium + FCS or sheep serum the cumulative release over 30/31 days was 11–34% for GDF5 and 6–17% for BB-1; the release of BMP-2 over 14 days was 25.7%. Addition of 10% PLGA fibers significantly augmented the 14-day release of GDF5 and BMP-2 (to 22.6% and 43.7%, respectively), but not of BB-1 (13.2%). All BMPs proved to be bioactive, as demonstrated by increased ALP activity in several cell lines, with partial enhancement by 10% PLGA fibers, and by a specific, early regulation of osteogenic/chondrogenic genes and proteins in hASCs. Between 10% and 45% of bioactive BMPs were released in vitro from CPC + PLGA fibers over a time period of 14 days, providing a basis for estimating and tailoring therapeutically effective doses for experimental and human in vivo studies.

## 1. Introduction

Substantial progress has recently been achieved concerning new synthetic biomaterials for bone repair and replacement, e.g., polymers, metals, ceramics, bioactive glasses, calcium sulfates, calcium carbonates, and calcium phosphates [1]. Also, novel mineral–organic tissue engineering scaffolds are currently designed to mimic the structure and molecular composition of native bone. Examples for the mineral components include calcium phosphate cement (CPC; [1,2]), bioceramics [3], biosilicates [4], graphene oxides [5], and clay nanotubes [6], whereas examples of organic component are poly(l-lactide-*co*-glycolide) acid (PLGA; [3,4,5]), carboxylmethylcellulose, and hyaluronan-bisphosphonate [2] or chitosan–gelatine–agarose hydrogels [1,6]. On the basis of promising previous in vitro and in vivo studies with a biodegradable, brushite-forming CPC with reinforcing PLGA fibers [7,8,9,10,11,12], the present study is focused on this particular mineral–organic bone replacement composite.

Calcium phosphate cement (CPC), first described in the 1980s [13,14], represents a potential alternative to polymethylmethacrylate (PMMA) for the surgical treatment of osteoporotic vertebral compression fractures, due to its biodegradability, fast setting ability, and osteointegrative capacity [15]. In addition, CPC significantly decreases the cement extrusion into the bone marrow compared to PMMA after both ex vivo and in vivo injections [16], possibly reducing the vertebral cement leakage and thus some serious side effects of kyphoplasty [17]. Also, CPC does not induce relevant tissue damage or inflammatory infiltration [9,18], whereas the high polymerization temperatures and low biocompatibility of PMMA may cause tissue necrosis and toxic reactions to PMMA monomers [19]. Furthermore, the elastic modulus of CPC (180 MPa, similar to that of cancellous bone) may avoid stress shielding effects and abnormal load transfer [20], thus potentially reducing secondary fractures of adjacent vertebral bodies due to the much higher elastic modulus of PMMA (2700 MPa). Finally, clinical efficacy of vertebroplasty/kyphoplasty has been reported for both PMMA cement [21] and CPC [22,23], although for CPCs there are some concerns on the long-term loss of the correction of the vertebral kyphosis angle [24,25].

In terms of CPC modifications, recent own studies have demonstrated that the inclusion of poly (l-lactide-*co*-glycolide) acid (PLGA) fibers in a brushite-forming CPC on one hand considerably improves its mechanical strength and fracture toughness ([26,27]; including scanning electron microscopy of the CPC), and on the other hand enhances in vitro osteogenic differentiation [7,8] and in vivo bone formation [9].

To further stimulate bone formation, the CPC can be used as a carrier for osteoinductive growth factors, e.g., bone morphogenetic proteins (BMPs; [10,11,28]). BMPs belong to the 33 member superfamily of transforming growth factor-β (TGF-β) proteins [29]. Their activity was first described around 1970, when their capacity to induce ectopic bone formation in adult animals was discovered [30,31]. After cloning and further characterization until 1990 [32,33], the central role of BMPs in bone or cartilage formation and osteogenic cell differentiation has been well documented [34]. The therapeutic efficacy of several BMPs has been tested in animal models of osteopenia. Local or systemic administration of BMP-2 improves bone structure or bone formation in mice [35], rats [36], goats [37], and sheep [38], and BMP-2 is a widely used molecule for lumbar spine fusion [39]. Also, systemic administration of rhBMP-6 increases bone volume and mechanical stability of bone in aged ovariectomized rats [40]. Likewise, local delivery of rhBMP-7 to vertebral bodies in osteopenic sheep improves bone mechanical strength and histomorphometric parameters [41]. This has led to the clinical use of products containing rhBMP-2 for fracture treatment or spinal fusion, although some safety concerns have arisen [42].

Recently, GDF-5 (also referred to as BMP-14) has shown positive effects in a sheep model of lumbar osteopenia by enhancing middle to long-term bone formation via changes in bone structure, formation, resorption, and compressive strength [10].

A mutant GDF5 protein called BB-1, which carries two point mutations in the BMP type1 receptor (BMPR-I) binding site of GDF-5 (and thus shows increased affinity to the subunit BMP receptor type 1A (BMPR-IA) [43]), also significantly enhanced the bone formation in a sheep lumbar osteopenia model for at least 3 months after a single therapeutic application [11].

However, despite promising in vivo effects of GDF5 and BB-1 [10,11], little is known on the quantity and bioactivity of these BMPs released from CPC or other bone replacement materials. Some in vitro release studies have focused on the BMP-2 release from surface-coated CPC [44,45], modified CPC [18,37,45,46,47,48] or other carriers such as chitosan scaffolds [49,50] or collagen-based CP composites [51], whereas only two studies have thus far investigated the release of GDF5 from collagen membranes [52] or hydrogels [53], but not from CPC. Finally, to our knowledge no data have been published on the release of BB-1 from bone replacement materials.

The present study sought to quantify the in vitro release and bioactivity of different BMPs from CPC formulations successfully used in sheep lumbar osteopenia models [10,11,12]. The BMPs GDF5, BB-1, and BMP-2 were chosen on the basis of their known clinical use or potential clinical relevance. The different BMP dosages were chosen based on therapeutic ranges previously used in large animal models [10,11,12].

## 2. Materials and Methods

### 2.1. Fabrication of PLGA Fibers and PLGA Fiber-Reinforced Cement

PLGA fibers were prepared from the granulate material PURASORB PLG 1017 (10/90 molar ratio of l-lactide and glycolide; Purac, Gorinchem, Netherlands) using a mini extrusion system (RANDCASTLE EXTRUSION SYSTEMS INC, Cedar Grove, NJ, USA; resulting diameter 25 µm) and cut to 1 mm length using a cutting mill PULVERISETTE 19 (FRITSCH GmbH, Idar-Oberstein, Germany) [26]. The commercially available brushite-forming CPC JectOS+ (Conformité Européenne (CE)-certified; Kasios, L’Union, France) was used. JectOS+ consisted of β-tricalcium phosphate (β-TCP, 98.5% *w*/*w*) and 1.5% *w*/*w* tetrasodium pyrophosphate and was mixed according to the manufacturer’s instructions.

CPC powders with different fiber content (5% or 10% (*w*/*w*)) were produced by mixing defined amounts of fibers und CPC powder in pure isopropanol with a high shear stirrer. Afterwards, the isopropanol was removed by evaporation. The powder-to-liquid ratio was 2.2 for all experiments.

### 2.2. Production of Recombinant GDF5, BB-1, and BMP-2

The recombinant human (rh) growth factors GDF5 (also called BMP-14), BB-1, and BMP-2 were produced in *E. coli* by the company BIOPHARM GmbH (Heidelberg, Germany; GDF5, BB-1) and the Hans-Knöll-Institut in Jena (BMP-2) using patented procedures [54]. The material was analyzed for endotoxin and total DNA content using established procedures (LAL test; [55]). The results revealed very low contamination for endotoxin (rhBMP-2: <0.11 IU/mg; rhGDF5: <0.17 IU/mg) and DNA content (rhBMP-2: <0.01 pg/µg), both well below the threshold values for therapeutic products (https://www.fda.gov/downloads/drugs/guidances/ucm310098). The GDF5 mutant BB-1 was produced via site-directed mutagenesis, as previously described [43].

### 2.3. Release of Low-Dose GDF5 and BB-1 from the CPC

CPC was used to obtain cement discs with a radius and a height of 2.0 mm, and a corresponding volume of 25.12 µL (mm^3^). To form the growth factor-loaded cement paste, 2 µg/mL and 10 µg/mL of the lyophilized growth factors GDF5 and BB-1 were dissolved in the liquid phase (in analogy to [26]) and then thoroughly manually mixed with the β-TCP/tetrasodium pyrophosphate cement powder (see above). The CPC paste was then filled into the respective molds and allowed to self-set and harden in situ. The BMP dosages were chosen because they reflect the low-dose BMP groups in own sheep in vivo experiments (1 and 5 µg in 500 µL each; [10,11,12]). The discs were initially pre-washed six times in DPBS to remove harmful components [7], and then either left non-ground or thoroughly ground with a glass rod in 1 mL of either PBS or cell culture medium (alpha-MEM; Gibco™, Life Technologies, Darmstadt, Germany) containing 10% fetal calf serum (FCS; Gibco™, Life Technologies). The release of GDF5 and BB-1 from non-ground (control) or ground cement discs at 37 °C was then measured at 1 h, as well as 1, 2, 3, 6, 8, 10, 13, 15, 17, 20, 22, 28, and 30 or 31 days using an ELISA assay developed by the company Biopharm GmbH, in which the specific anti-GDF5 and anti-BB-1 antibodies recognize only correctly folded, presumably bioactive proteins.

### 2.4. Release of High-Dose GDF5 and BB-1 from the CPC

Cement cuboids (length 10 mm, height and width 5.0 mm) with a corresponding volume of 250 µL (mm^3^) were prepared from JectOS+ CPC. During the preparation of the cement cuboids, 200 µg/mL and 1000 µg/mL of the growth factors GDF5 and BB-1 were loaded in the CPC formulation. These dosages are representative of the high-dose BMP groups in our published sheep in vivo experiments (100 and 500 µg in 500 µL each; [10,11,12]). The samples underwent the same pre-washing and grinding procedures as described above and were then incubated at 37 °C in 2 mL of either PBS or sheep serum. The release of GDF5 and BB-1 from non-ground (control) or ground cuboids was measured at 1 h, and 1, 2, 5, 7, 9, 12, 14, 16, 19, 22, 27, and 30 days using an ELISA (see above).

### 2.5. Release of GDF5, BB-1, and BMP-2 from PLGA Fiber-Reinforced CPC

The PLGA fiber-reinforced CPC (5 and 10% fibers; *w*/*w*) was used to obtain discs with a radius of 4 mm, a height of 0.5 mm, and a corresponding volume of 25.12 µL (mm^3^), which were loaded with an intermediate high dose of 400 µg/mL GDF5, BB-1, and BMP-2.

Samples consisting of pure CPC (controls) or CPC ± PLGA fibers were also pre-washed and ground as above and then incubated at 37 °C in 1 mL of sheep serum. The release of GDF5, BB-1, and BMP-2 from non-ground (control) or ground cement discs was measured at 1 h, as well as 1, 3, 7, and 14 days using a self-developed ELISA (see above) for GDF5 and BB-1, and a commercial ELISA for BMP-2 (Quantikine, R&D Systems, Minneapolis, MN, USA).

Data for all release experiments were expressed as: (i) time-dependent absolute BMP release in ng/mL; (ii) cumulative release in ng over up to 31 days; (iii) release in % of the applied dosage; and (iv) % retention (applied dosage—released dosage/applied dosage in %).

### 2.6. Alkaline Phosphatase (ALP) Activity of the C2C12, C2C12BRlb, MCHT -1/26, and ATDC-5 Cell Lines Following Exposure to the Extracts of BMP-Loaded, PLGA Fiber-Reinforced CPC

PLGA fiber-reinforced CPC discs (geometry see Section 2.5) were loaded with 400 µg/mL GDF5, BB-1 or BMP-2 and ground. BMPs were extracted at 37 °C in 1 mL of sheep serum for 3 days. Then 100 µL of the extracts were added to 200 µL of culture medium (DMEM + 10% FCS, Gold (Seraglob, Schaffhausen, Switzerland) + l-Glutamine for C2C12 (ATCC/LGC Standards GmbH, Wesel, Germany; kindly provided by Prof. Dr. Petra Knaus; Institute of Chemistry and Biochemistry; Free University Berlin, Berlin, Germany) and C2C12BRlb (BIOPHARM GmbH; with overexpressed BMP-receptor-1b); DMEM + 10% FCS, Gold for MCHT-1/26 (BIOPHARM GmbH); DMEM/F12 10% FCS, Gold for ATDC-5 (RIKEN BioResource Research Center, Tsukuba, Ibaraki, Japan; BIOPHARM GmbH). Cell lines were cultured for 3 days in the modified media in 48-well plates at 37 °C, 5% CO_2,_ and a density of 4.5 × 10^3^ cells/well.

To investigate the biological activity of the extracts, the cells were washed once with PBS and lysed (1% (*v*/*v*) Nonidet 40 (Sigma, Taufkirchen, Germany); 0.1 M glycine; 1 mM MgCl_2_). A substrate solution was prepared containing 4 mL of diethanolamine substrate buffer (5%; *v*/*v*), 0.148 g/10 mL *p*-nitrophenyl phosphate (pNPP; both Thermo Scientific, Rockford, IL, USA), and 6 mL of distilled water. Seventy µL of the cell lysate and 70 µL of the substrate solution were applied to each well (96-well plate). The ALP activity in each sample was measured at different time points with an ELISA reader at 405 nm (Tecan, Männedorf, Switzerland). For each cell line, the control group was represented by cells cultured with the extracts of growth factor-free CPC.

### 2.7. Isolation of Human Adipose Tissue-Derived Mesenchymal Stem Cells (hASCs)

hASCs were isolated using a well-established method described previously ([8] and references therein). For cell isolation, subcutaneous adipose tissue was collected from both male and female subjects (*n* = 8, mean age 39.8 ± 4.9 years). The study was approved by the ethics committee of the Jena University Hospital (approval registration number: 3331-21/11; date of approval 30 January 2012) and all donors gave written consent prior to the procedure. The tissue was washed six times with an equal volume of pre-warmed PBS with penicillin/streptomycin to remove blood components. An equal volume of pre-warmed collagenase solution (0.1% type I collagenase (Roche, Mannheim, Germany) and 1% bovine serum albumin dissolved in PBS supplemented with 2 mM calcium chloride) was then added to the tissue samples and incubated at 37 °C for 60 min. After the collagenase digestion, the sample was spun down at 300× *g* at room temperature (RT) for 5 min. For the disaggregation of stromal cells from primary adipocytes, the sample was shaken vigorously to disrupt the pellet and to mix the cells. Thereafter, the sample was spun down again at 300× *g* at RT for 5 min. The top layer of fat, oil, and primary adipocytes and the underlying collagenase solution was carefully removed. The hASC pellet was re-suspended in PBS and spun down again at 300× *g* at RT for 5 min. The supernatant was removed, the cells were suspended in DMEM/F12 with 10% FCS, 1% Gentamycin (10 mg/mL), and 1% Penicillin/Streptomycin (10,000 Units/mL, 10 mg/mL, respectively; all Invitrogen, Darmstadt, Germany) and cultured in 225 cm^2^ flasks for 7 days (1 × 10^7^ cells/flask; medium change every 2 days). Thereafter, hASCs were trypsinized and characterized by flow cytometry ([8]; mesenchymal stem cell markers: CD29: 85.4% ± 3.5%; CD44: 86.5% ± 2.3%; CD73: 57.8% ± 6.2%; CD90: 94.7% ± 1.2%; CD105: 77.2% ± 9.0%; markers of monocytes, leukocytes, and endothelial cells: CD14: 2.6% ± 0.4%; CD45: 11.0% ± 2.9%; CD31: 8.1% ± 1.5%, respectively). However, since 30.9% ± 7.4% of the cells expressed the hematopoietic progenitor cell antigen CD34 on their surface, they were subjected to anti-CD34 negative purification with Dynabeads^®^ CD34 (Invitrogen; [8] and references therein) to remove CD34-positive hematopoietic progenitor cells. This anti-CD34 negative purification reduced the proportion of CD34-positive cells to 2.9% ± 1.3%.

### 2.8. Extraction of GDF5 from the CPC, Exposure of hASCS to the Extracts, RNA Isolation, cDNA Synthesis, and RT-PCR

CPC discs (geometry as in Section 2.5), PLGA fiber-reinforced CPC discs, and fiber-reinforced CPC discs with 20 µg/mL or 200 µg/mL GDF5 were initially pre-washed six times in DPBS to remove harmful components and then ground with a glass rod. Thereafter, 1 mL pooled sheep serum was added for 3 days at 37 °C. Then 100 µL of the extracts were added to 200 µL of culture medium.

For gene expression analysis, 1 × 10^5^ hASCs were seeded in 12-well plates and exposed for 3 days to the above-described extracts of CPC discs without fibers and with 10% (*w*/*w*) fiber content, the latter either without GDF5 or doped with low dose (20 µg/mL) or high dose GDF5 (200 µg/mL).

Thereafter, total RNA was isolated from the hASCs by adding the lysis buffer component of a commercial RNA isolation kit (Macherey & Nagel, Düren, Germany) directly to the CPC discs and then reverse-transcribed as previously described ([8] and references therein). mRNA expression of the osteogenic markers runt-related transcription factor 2 (Runx2), osterix, alkaline phosphatase, collagen 1, osteopontin, and osteocalcin, the chondrogenic markers collagen 2 and aggrecan, as well as the house-keeping gene GAPDH was analyzed by real-time PCR using a RealPlex^®^ PCR machine (Eppendorf, Hamburg, Germany). Primer pairs and PCR conditions are shown in Table 1. The relative mRNA concentrations of the analyzed genes in each sample were calculated using an external standard curve and the ΔΔ*C*t method. Product specificity of the real-time PCR was validated by: (i) melting curve analysis; (ii) agarose gel electrophoresis; and (iii) initial cycle sequencing of the PCR products.

### 2.9. Protein Extraction from hASCs and Enzyme-Linked Immunosorbent Assay (ELISA)

For protein analysis, 2 × 10^5^ hASCs were seeded in 6-well plates and exposed for 3 days to the above-described extracts of CPC discs without fibers and with 10% (*w*/*w*) fiber content, the latter either without GDF5 or doped with low dose (20 µg/mL) or high dose GDF5 (200 µg/mL).

At the end of the incubation time, cells were washed twice with ice-cold phosphate-buffered saline (PBS), incubated for 15 min with buffer for protein extraction (50 mM Tris, 150 mM NaCl, EDTA, pH 7.4, containing 100 mM NP40, 1 mM phenylmethylsulphonylfluoride, 1 mM Na_3_VO_4_,) and stored in a tube at −80 °C for subsequent analysis.

Collagen 1 concentrations in cell lysates were then quantified according to the protocol of a commercial ELISA kit (Chondrex, Redmond, WA, USA; BlueGene, Shanghai, China). Absorption was measured at 490 nm using a Fluostar Optima Reader (BMG Labtech GmbH, Offenburg, Germany).

### 2.10. Statistical Analysis

The data were expressed as means ± SEM. Significance was tested using the non-parametric Kruskal–Wallis and Mann–Whitney U tests and the IBM SPSS Statistics 21 program. Differences were considered statistically significant for *p* ≤ 0.05.

## 3. Results

### 3.1. In Vitro Release of GDF5 from the CPC

There was hardly any release of GDF5 from the non-ground CPC (in PBS or medium with 10% FCS/sheep serum) or from the ground CPC in PBS, irrespective of the GDF5 doses used (< 0.1% of the loaded dose; data not shown).

The release of GDF5 from the ground CPC incubated in cell culture medium with 10% FCS showed a typical kinetics, characterized by an initial burst/peak release until day 3 to 6 for all doses (2, 10, 200 and 1000 µg/mL of GDF5), followed by a slower increase thereafter (for details see Figure 1A and Figure 2A).

The cumulative release curves showed a rapid increase to 16.5 ng and 43.8 ng within 3 days in the case of the low GDF5 doses (2 and 10 µg/mL respectively; Figure 1B), pointing out that 31% and 18% of the respective low doses were initially released (Figure 1C), resulting in a retention of 69% and 82% (Figure 1D). The same trend was observed in the case of the high GDF5 doses (200 and 1000 µg/mL) with a rapid increase to 2961 ng and 15,476 ng within 2 days (200 and 1000 µg/mL respectively; Figure 2B), indicating a release of only 6% and 7% of the respective total doses (Figure 2C) and a retention of 94% and 93% (Figure 2D).

For all doses, the cumulative release then slightly and progressively increased to reach a plateau, suggesting a continuous, long-term release of the protein. At day 30, 17.3, and 61.4 ng of the initial low doses of GDF5 were released (2 and 10 µg/mL respectively; Figure 1B), resulting in a final release percentage of 34% and 25% (retention of 66% and 75% Figure 1C,D). Concerning the high GDF5 doses, 5324 and 32,194 ng were released in total over 30 days (200 and 1000 µg/mL respectively; Figure 2B), with a final release percentage of 11% and 13% (retention of 89% and 87%; Figure 2C,D).

### 3.2. In Vitro Release of BB-1 from the CPC

As in the case of GDF5, there was hardly any BB-1 release from either the non-ground CPC (in PBS or medium with 10% FCS/sheep serum) or from the ground CPC in PBS for any of the analyzed BB-1 doses (< 0.1% of the loaded dose; data not shown).

The release of BB-1 showed kinetics comparable to the ones of GDF5, characterized by an early burst release and a subsequent continuous, slow release (for details see Figure 1E and Figure 2E).

The cumulative release curves showed a rapid increase to 2 ng and 29.0 ng within 3 days in the case of the low BB-1 doses (2 and 10 µg/mL respectively; Figure 1F). Therefore, 5% and 12% of the respective low doses were initially released (Figure 1G), resulting in a retention of 95% and 88% (Figure 1H). A similar trend was observed also in the case of the high BB-1 doses, as the release rapidly increased to 1248 and 9370 ng at day 1 (200 and 1000 µg/mL respectively; Figure 2F), showing a release of only 4% for both concentrations within 2 days (Figure 2G) and a retention of 96% (Figure 2H).

The cumulative release then continued to increase slowly, reaching a plateau with final release values of 2.7 and 41.7 ng at day 31 for the low doses (2 and 10 µg/mL respectively, Figure 1F), as well as 3441 and 21,271 ng at day 30 for the high doses (200 and 1000 µg/mL respectively, Figure 2F). Therefore, 6% and 17% of the respective initial low doses (retention of 94% and 83%; Figure 1G,H) and 7% and 9% of the respective high doses were finally released (retention of 93% and 91%; Figure 2G,H).

### 3.3. Influence of PLGA Fibers on the In Vitro Release of BMPs from the CPC

#### 3.3.1. GDF5

The release of GDF5 (400 µg/mL) was increased by the presence of 5% PLGA fibers and, in particular, 10% PLGA fibers (peak release increased from 968 ng/mL at 1 h to 1157 and 1293 ng/mL, respectively; Figure 3A; Table 2).

Accordingly, the cumulative release of GDF5 was significantly increased by the presence of 5% PLGA fibers and, in particular, 10% PLGA fibers (*p* ≤ 0.05 for 10% PLGA versus pure CPC at all time points; pure CPC: max. 1764 ng; 5% fibers: 1854 ng; 10% fibers: 2259 ng; Figure 3B); also, the % release within 14 days was significantly increased (*p* ≤ 0.05 for 10% PLGA versus pure CPC at all time points; pure CPC: max. 18%; 5% fibers: 19%; 10% fibers: 23%; Figure 3C; for the respective retention values see Figure 3D).

#### 3.3.2. BB-1

The release of BB1 (400 µg/mL) was only marginally influenced by the presence of 5% or 10% PLGA fibers (peak release of 536, 531 and 622 ng/mL, respectively; Figure 3E; Table 2).

This was confirmed by the cumulative release (pure CPC: max. 1129 ng/mL; 5% fibers: 1071 ng/mL; 10% fibers: 1381 ng/mL; Figure 3F) and the % release at 14 days (pure CPC: max. 11%; 5% fibers: 11%; 10% fibers: 13%; Figure 3G; for the respective retention values see Figure 3H).

At selected time points, the BB-1 release from pure CPC and/or CPC ± PLGA fibers was significantly lower than the GDF5 release (see Table 2 for the absolute and % release).

#### 3.3.3. BMP-2

The release of BMP-2 (400 µg/mL) was markedly and significantly increased by the presence of 5% PLGA fibers and, in particular, 10% PLA fibers (Figure 3I; Table 2).

This was also reflected in a significantly boosted cumulative release (pure CPC: max. 2571 ng/mL; 5% fibers: 3392 ng/mL; 10% fibers: 4372 ng/mL; Figure 3J) and an increased % release within 14 days (pure CPC: max. 26%; 5% fibers: 34%; 10% fibers: 44%; Figure 3K; for the respective retention values see Figure 3L).

Except for the early time point 1 h (BMP-2 release < GDF5 release), the BMP-2 release from pure CPC and CPC ± PLGA fibers was always significantly higher than the release of GDF5 and BB-1 throughout the whole period of 14 days (see Table 2).

### 3.4. Bioactivity of the BMPs Released from the CPC

#### 3.4.1. Fold-Change Effects of GDF5, BB-1, and BMP-2 Extracts on the ALP Activity in the Cell Line C2C12

Whereas the 3 day extracts of GDF5- or BMP-2-loaded, ground CPC ± PLGA fibers did not induce any ALP activity in C2C12 cells, the extracts of BB-1-loaded CPC induced a higher ALP signal than the extracts of BB-1-free CPC (1.8-fold induction versus CPC without growth factor at 30 min; 2.3-fold at 60 min), an effect that was further enhanced by the presence of 10% PLGA fibers (to 11.9-fold at 30 min; Figure 4).

#### 3.4.2. Fold-Change Effects of GDF5, BB-1, and BMP-2 Extracts on the ALP Activity in the Cell Line C2C12BRIb

In contrast to the effects on the cell line C2C12, the extracts of GDF5, BB-1, or BMP-2-loaded, ground CPC ± PLGA fibers all induced an ALP activity in BMPR1B receptor-transfected C2C12BRIb cells (5.2-fold, 4.4-fold, and 4.8-fold, for GDF5, BB-1, or BMP-2, respectively, at 60 min; Figure 5). Whereas this effect was further enhanced by the presence of 10% PLGA fibers in the case of GDF5 and BB-1 extracts (to 10.3-fold and 18.3-fold, respectively at 30 min), there was no further enhancement in the case of BMP-2 extracts (Figure 5).

#### 3.4.3. Fold-Change Effects of GDF5, BB-1, and BMP-2 Extracts on the ALP Activity in the Cell Line MCHT-1/26

Also in MCHT-1/26 cells, GDF5, BB-1, or BMP-2-loaded, ground CPC ± PLGA fibers all induced an ALP activity (9.3-fold at 60 min for GDF5, 46.3-fold at 30 min for BB-1, and 1.2-fold at 60 min for BMP-2; Figure 6). This effect was further enhanced by the presence of 10% PLGA fibers in the case of BMP-2 extracts (to 2.5-fold at 60 min; Figure 6).

#### 3.4.4. Fold-Change Effects of GDF5, BB-1, and BMP-2 Extracts on the ALP Activity in the Cell Line ATDC5

In ATDC5 cells, finally, GDF5, BB-1, or BMP-2-loaded, ground CPC ± PLGA fibers all induced an ALP activity (7.9-fold at 5 min for GDF5, 68-fold at 10 min for BB-1, and 1.4-fold at 5 min for BMP-2; Figure 7). This effect was further enhanced by the presence of 10% PLGA fibers in the case of BMP-2 extracts (to 2.2-fold at 5 min; Figure 7).

#### 3.4.5. Effects of GDF5 Extracts on the Gene Expression in hASCs

In comparison to their expression in hASCs exposed for 3 days to the extracts of pure CPC, the mRNA expression of the osteogenic transcription factors Runx2 and osterix, the osteogenic markers ALP, type I collagen, osteopontin, and osteocalcin, as well as the chondrogenic markers type II collagen and aggrecan, were in all cases significantly upregulated by exposure to the extracts of CPC containing 10% PLGA fibers (CPC+F; 4- to 364-fold; *p* ≤ 0.05; Figure 8A–H). Somewhat surprisingly, however, exposure to the extracts of CPC containing 10% PLGA fibers and either low dose (10 µg/mL; CPC+F+G) or high dose GDF5 (200 µg/mL; CPC+F+hG) numerically or significantly downregulated the mRNA expression of these genes when compared to CPC+F (*p* ≤ 0.05 for Runx2 -low dose-; and osterix, ALP, osteopontin, osteocalcin, type II collagen, and aggrecan –both doses). In the case of Runx2 (low dose) and type I collagen (high dose) these values still remained significantly higher than those for pure CPC, in the case of osteopontin (high dose) the values were significantly lower than those for pure CPC (Figure 8A–H).

#### 3.4.6. Effects of GDF5, BB-1, and BMP-2 Extracts on the Protein Expression in hASCs

In contrast to the mRNA expression, the protein expression of type I collagen was numerically downregulated by 3-day exposure to the extracts of CPC containing 10% PLGA fibers (CPC+F), when compared to the extracts of pure CPC (CPC; 4-fold; Figure 9). In this case, addition of low dose GDF5 (CPC+F+G; 2.2-fold; *p* ≤ 0.05) or high dose GDF5 (CPC+F+hG; 1.6-fold) upregulated the protein expression of type I collagen in comparison to CPC+F (Figure 9).

## 4. Discussion

This in vitro study sought to analyze the quantity and the bioactivity of different BMPs released from a biodegradable, brushite-forming CPC, the latter successfully used in vivo for bone regeneration in a sheep model of lumbar osteopenia [10,11,12]. The results indicated that: (i) the cumulative release in medium + FCS/sheep serum from the CPC within 30/31 days reached 11–34% for GDF5 and 6–17% for BB-1. For BMP-2, the release within 14 days was 25.7%; (ii) addition of 10% PLGA fibers significantly augmented the 14-day release of GDF5 and BMP-2 (to 22.6% and 43.7%, respectively), but not of BB-1 (13.2%); and (iii) the released BMPs were bioactive, as shown by increased ALP activity in different cell lines, in some cases further augmented by the presence of 10% PLGA fibers. Bioactivity of the released BMPs was further confirmed by specific, early regulation of osteogenic/chondrogenic genes and proteins in hASCs.

Considerable amounts of bioactive BMPs were thus released from the present CPC, which seems to qualify as a suitable drug delivery system for BMPs in bone pathology [18,28]. A similarly broad range of BMP release (15% and 38%) has also been observed in previous studies [37,46] reporting on modified CPC with comparable dosages of BMP-2 (in the second study in comparison to TGF-β1).

Regarding the enhancement of BMP release by the presence of PLGA fibers, possible explanations include a high solubility of the PLGA fibers in physiological fluids and increased affinity of the BMPs for the fibers [28,50,56,57], although there are currently no data on the BMP release from CPC-PLGA fiber composites. The enhancement of the BMP release is thus another interesting feature of PLGA fibers, in addition to their known effects on the mechanical stability [26,27], degradability [58], and increased osteoconductivity of the CPC in a sheep lumbar vertebroplasty model [9]. Since the PLGA fibers applied in the present study (10:90 molar ratio of l-lactide and glycolide) were chosen on the basis of their established clinical use as suture material, there is currently no information on differential effects of the l-lactide and glycolide components on the release of the different BMPs [59].

### 4.1. Kinetics of BMP Release

The release pattern of all three BMPs was similar—i.e., there was an initial burst release usually around 1 day—followed in some cases by a second peak between day 6 and day 9 (especially at higher BMP doses). These profiles are comparable to those observed when BMP-2 was mixed with demineralized bone putty (DBM; [60]), and appear to consist of an early bulk release from a “loose” compartment within the first 2–3 days, followed by slower release from a more tightly packed second compartment [28,44,60,61,62]. In another study, however, a burst release of BMP-2 was only observed when the BMP-2 was surface-adsorbed onto a preset CPC, but not when incorporated into the CPC by adding it to the liquid component of the CPC [46], indicating a relevant role of the particular features of the carrier material [46,60]. Notably, for higher BMP doses (≥ 200 µg/mL), the cumulative release of the three different BMPs reached levels > 500 ng already after 1 day—i.e., levels sufficient to stimulate osteoblastic differentiation and proliferation of mesenchymal stem cell lines in vitro [63]—a result potentially relevant for the induction of bone healing in vivo [10,11,12].

### 4.2. Differences among the Three BMPs

The present study confirms the considerable differences in the release of different members of the TGF-beta superfamily, emphasizing the need to characterize the influence of the individual features of each protein on its binding to the CPC and the subsequent release [37,46,52,53,61,64].

In particular, BMP-2 shows a very high affinity for calcium phosphates, which appears to be driven by chemical interactions between the hydroxyl, amine, and carboxyl groups in BMP-2 and the divalent Ca^2+^ ions present in the carrier [28,61,65,66]. Another important factor is the 3D arrangement of the BMP molecule, possibly including the binding regions for the BMP receptors [67,68,69]. In view of potential therapeutic applications, carrier, dose, and route of administration thus need to be carefully established for every BMP, with in vitro and in vivo pharmacokinetic validation [45,67].

At the same time, the release of different BMPs also depends on the particular properties of the carrier material, such as the individual composition or spatial organization of the CP components and their physicochemical features (e.g., surface coating with soluble, bioactive or nanocrystalline CPs; local pH; microporosity [7,44,45,64,65]). The JectOS+ cement used in the present study has a porosity of 40%, with a major proportion of small pores (diameter approximately 1 µm) and a low proportion of large pores (diameter approximately 200 µm; Kasios, technical file), as also confirmed by own micro-CT analysis (unpublished). While macropores and micropores differentially affect cell immigration and angiogenesis versus nutrient transport and bone integration [70], a differential influence on the release of the three BMPs is presently unclear.

To our knowledge, only two studies have directly addressed the release of GDF5 from bone replacement materials, i.e., from collagen membranes [52] or from photo-cured hyaluronic acid hydrogels [53]. While in the first study the release of GDF5 was not quantified, the release of GDF5 from photo-cured hyaluronic acid hydrogels over a period of 28 days was always > 70% for doses between 10 and 1000 ng/mL. However, these results are difficult to compare with the present GDF5 release data, since the carrier materials are considerably different. Finally, to our knowledge, there is no published study on the release of BB-1 from bone replacement materials.

### 4.3. Bioactivity of the Released BMPs

Although the CPC used in the present study (brushite-forming JectOS+) reaches a pH below 2.0 for a few minutes during the curing process and then continues to harden at a pH of approximately 4.0 (Kasios, technical file), the BMPs released from the CPC remained at least partially osteogenic for four different marker cell lines. When comparing the amounts of BMP released within 3 days from the CPC (see Table 2) to the standard dilution curve of the BMPs in ALP bioactivity assays, maximal recovery rates of 31.8%, 36.0%, and 12.3% were estimated for bioactive GDF5, BB1, and BMP-2, respectively. This suggests that a considerable proportion of the released BMP is bioactive.

This is in agreement with the known stability of BMPs in mildly acidic buffers and calcifying matrix vesicles, in which there is abundant release of protons during hydroxyapatite formation [71,72,73]. Because of the low BMP-2 solubility at pH values above 6 [74,75], these mildly acidic conditions (pH of 4.5) are also used for the formulation of marketed BMP-2 products like INFUSE^®^.

To our knowledge, only some studies [45,47,48,56] have shown the release of bioactive BMPs from CPCs, and thus the present results further support the suitability of CPC as a drug delivery system.

The different BMPs analyzed in the present study showed differential induction of ALP activity in individual marker cell lines—i.e., GDF5 and BMP-2 reacted with all cell lines except for C2C12—whereas the GDF5 mutant BB-1 reacted with all cell lines. In addition, augmentation of ALP activity by the presence of 10% PLGA fibers was only detectable in selected cell lines. This degree of variability is expected given the known properties of the different cell lines, including differential type I BMPR expression (for example C2C12 and ATDC5 cells exclusively carry the BMPR-IA; [76,77,78,79,80]), with functional relevance for their osteogenic differentiation [81,82,83,84]; differential affinity of a given BMP for individual type I or type II BMPR [85]; differential functional sensitivity to PLGA itself or PLGA breakdown products [84,86]); and differential downstream signaling after activation of the BMPR-IA [87]). Hence, the present results underline that combinations of individual biomarker cell lines have to be selected for each given BMP and experimental question.

The bioactivity of the released GDF5 was confirmed by its specific influence on the gene and protein expression of osteogenic and chondrogenic markers in hASCs. Interestingly, gene expression of these markers was upregulated by the addition of 10% PLGA fibers to the CPC, but then downregulated by further addition of GDF5. This may be due to the early time point of GDF5 action on the hASCs (3 days), in line with previous results on the early influence of BMP-2–containing poly(l-lactic acid) nanofibers on growth/differentiation of human mesenchymal stem cells [59].

### 4.4. Long-Term Retention of a Depot of Therapeutically Applied BMP

The current CPC formulation, i.e., BMP incorporation into the body of the carrier, presumably leads to a homogeneous distribution throughout the CPC [28,45], with limited immediate release of BMP from the superficial loose compartment and long-term retention of therapeutic BMP for at least 30 days [58,67]. This may represent an advantage of the current CPC, in that this in vivo “depot” may guarantee sustained local release of BMP even after a one-time surgical application, and hence potential long-term beneficial effects on bone regeneration [10,11,12].

### 4.5. Limitations of the Present Study

Currently, the percentage of bioactive BMP released from the CPC can only be approximated by comparing the amounts of released BMP with the concentrations estimated from the ALP bioactivity assays. Although the current ELISA systems for GDF5 and BB-1 were already designed to recognize only correctly folded protein (data not shown), more precise information is expected from the use of marker cells transfected with BMP-reactive reporter gene constructs or receptor-ligand ELISAs designed to assess the binding of correctly folded BMP to its receptor(s). Since all ALP bioactivity assays included a standard dilution curve of the respective free, non-carrier bound BMPs (data not shown), an exemplary estimate of the concentrations of bioactive BMP released in 3 day extracts can be attempted for GDF5 (CPC ± 10% fibers loaded with 400 µg/mL; extract in 1 mL sheep serum) and the cell line C2C12BRIb (compare with Figure 5). In this case, the estimate results in maximally 160 ng/mL for CPC and maximally 640 ng/mL for CPC + 10% fibers. Considering the 3-fold dilution of the extracts for the ALP bioactivity assays (see Section 2.6.), final estimates of 480 ng bioactive GDF5 can be assumed for CPC and 1.920 ng for CPC + 10% fibers. These values are approaching, but are still somewhat lower than the values calculated from the cumulative release over 3 days for CPC (1639.58 ng) and CPC + 10% fibers (2188.71; compare with Figure 5 and Table 2), underlining the validity of the release ELISAs and the ALP bioactivity assays. However, the comparison is only partially valid, since the standard dilution curve is exclusively based on free, non-carrier bound GDF5, whereas the sheep serum extracts of the CPC discs may contain an unknown proportion of CPC particles, i.e., GDF5 in a carrier-bound form. Also, the present study provided release information limited to 1 month, thus the long-term BMP release from a second, more tightly packed ‘depot’ compartment of the CPC was not fully covered. Depending on the respective BMP and CPC formulation (± 5% or 10% PLGA fibers), this depot may regard between 56% and 94% of the BMP, in clear contrast to the lower retention of GDF5 in photo-cured hyaluronic acid hydrogels (max. 27%; 28 days; [53]) or on BMP-coated hydroxyapatite particles (max. 24%; 14 days; unpublished data).

In terms of relevance of the present in vitro results for in vivo applications, the present BMP concentrations were deliberately designed to reflect the low-dose and high-dose BMP used in own in vivo experiments in sheep analyzing the healing of bone defects after therapy with BMP-loaded CPC [10,11,12]. In particular, a total local dose between 5 and 100 µg GDF5 proved sufficient to significantly augment bone formation [10]. Assuming the higher dose (100 µg) for successful induction of bone formation in vivo, and considering a release of approximately 23% of the GDF5 from the CPC + 10% PLGA fibers within 14 days (Table 2), local doses as low as 23 µg may suffice for a therapeutic effect. These doses are in line with those therapeutically effective in sheep when applied on BMP-coated hydroxyapatite particles (5 µg and 50 µg; unpublished data) and considerably lower than dosages previously used in clinical applications (0.25 to 40 mg; [10] and references therein).

In general, in vitro experiments can never reflect the full range of cellular, enzymatic, and physicochemical factors acting at the local tissue level, and therefore the results need in vivo pharmacokinetic validation. As a matter of fact, a considerably increased in vivo BMP-2 release (10–20%) compared to the in vitro release has been previously reported from a BMP-2-loaded CPC in a subcutaneous rat model ([45] and references therein), suggesting that the present in vitro results may somewhat underestimate the in vivo BMP release.

## 5. Conclusions

The present study showed that: (i) considerable proportions of BMP were released from the CPC within 31 days; (ii) the presence of PLGA fibers significantly enhanced the BMP release within 14 days; and (iii) the released BMPs demonstrated bioactivity, in some cases augmented by the addition of 10% PLGA fibers. These data confirm that PLGA fiber-reinforced CPCs qualify as a suitable drug delivery system, releasing moderate amounts of bioactive BMPs sufficient to promote bone defect healing in large animal models [10,11,12]. The fiber-reinforced CPC may qualify for the treatment of compression fractures in load-bearing areas like vertebral bodies through minimally invasive vertebroplasty or kyphoplasty.

## Figures and Tables

**Figure 1 pharmaceutics-11-00455-f001:**
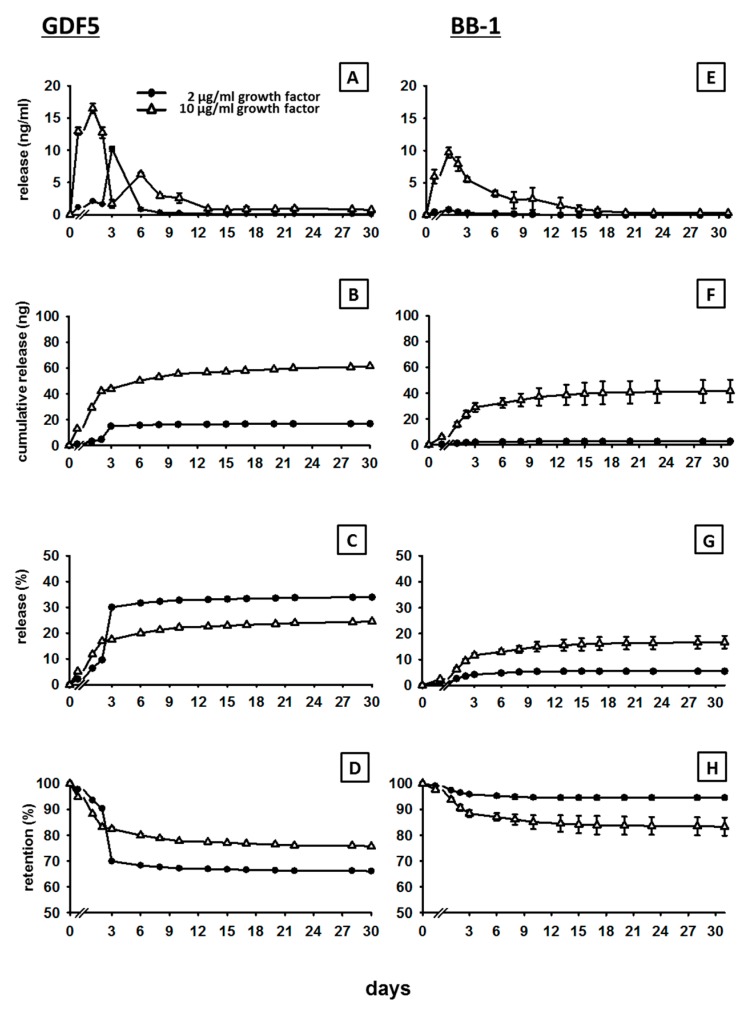
In vitro release of 2 and 10 µg/mL GDF5 and BB-1 from CPC. Cement discs (*r* = 2.0 mm, *h* = 2.0 mm, *V* = 25.12 µL) were loaded with 2 or 10 µg/mL GDF5 and BB-1, ground, and incubated at 37 °C in 1 mL of cell culture medium + 10% FCS for 1 h, as well as 1, 2, 3, 6, 8, 10, 13, 15, 17, 20, 22, 28, and 31 days. The release of GDF5 (**A**) and BB-1 (**E**) from ground cement discs was measured by ELISA and used for the calculation of cumulative release (**B**,**F**), % release (**C**,**G**), and retention (**D**,**H**; *n* = 2).

**Figure 2 pharmaceutics-11-00455-f002:**
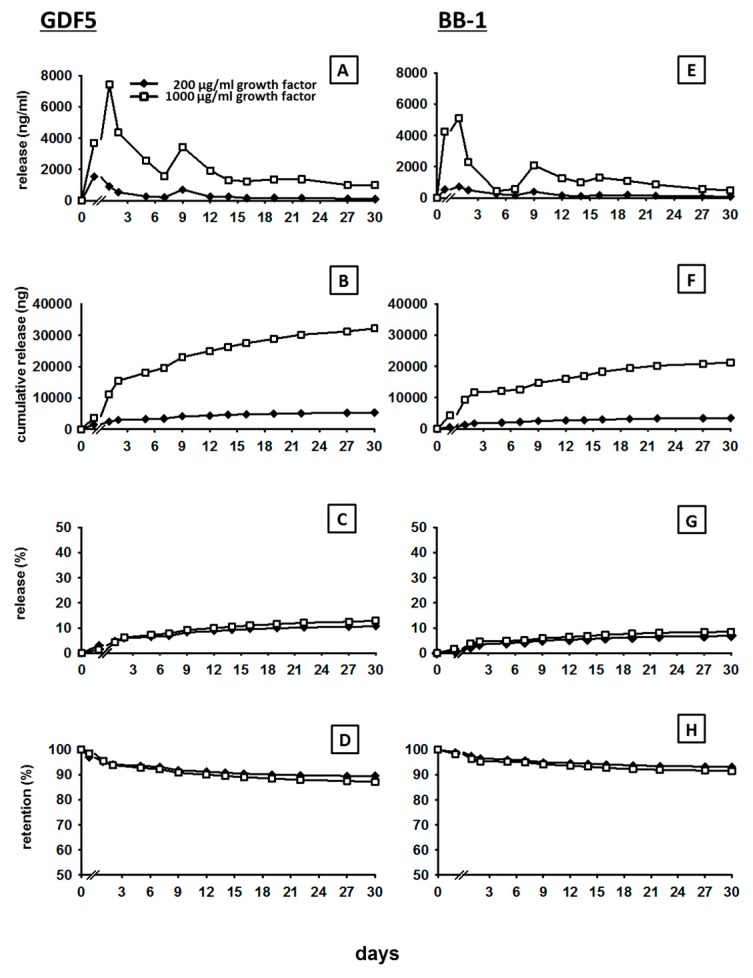
In vitro release of 200 and 1000 µg/mL GDF5 and BB-1 from CPC. Cement cuboids (*l* = 10 mm, *h* = 5.0 mm, *w* = 5.0 mm, *V* = 250 µL) were loaded with 200 or 1000 µg/mL GDF5 and BB-1, ground, and incubated at 37 °C in 2 mL of either PBS or sheep serum. After 1 h, as well as 1, 2, 5, 7, 9, 12, 14, 16, 19, 22, 27, and 30 days, the release of GDF5 (**A**) and BB-1 (**E**) from ground cement discs was measured by ELISA and used for the calculation of cumulative release (**B**,**F**), % release (**C**,**G**), and retention (**D**,**H**; *n* = 1).

**Figure 3 pharmaceutics-11-00455-f003:**
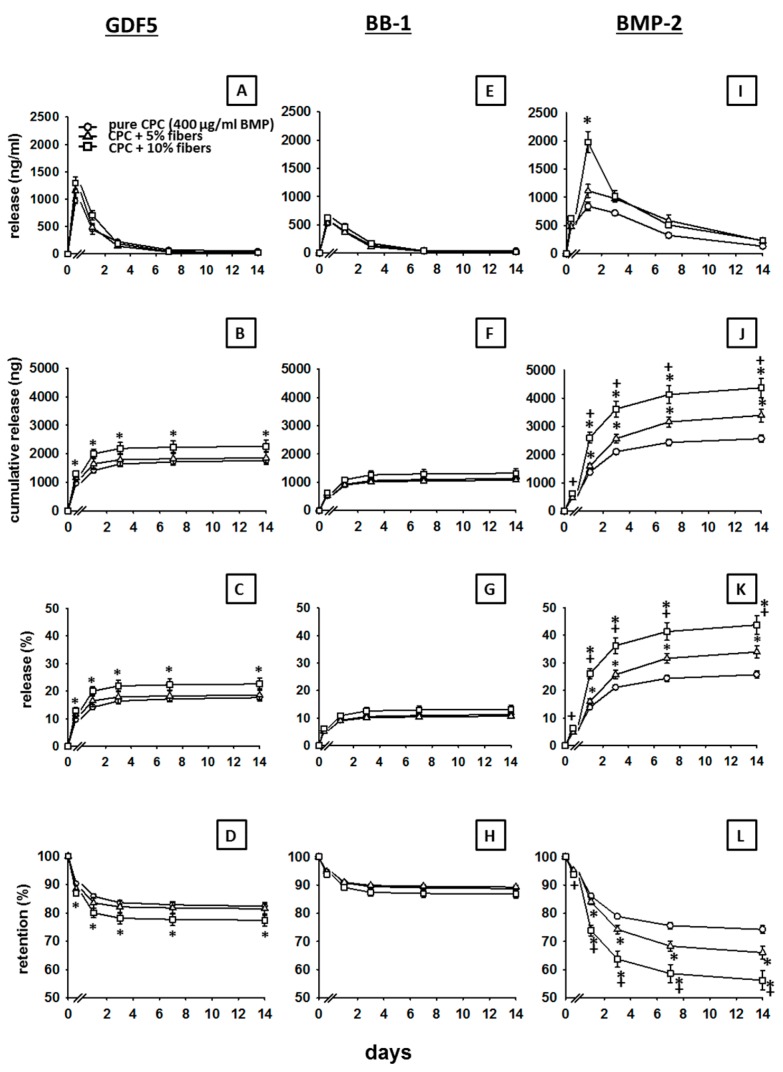
Release of GDF5, BB-1, and BMP-2 from PLGA fiber-reinforced CPC. Discs consisting of PLGA fiber-reinforced CPC (*r* = 4.0 mm, *h* = 0.5 mm, *V* = 25.12 µL) were loaded with 400 µg/mL GDF5, BB-1, and BMP-2, ground, and incubated at 37 °C in 1 mL of sheep serum. After 1 h, as well as 1, 3, 7, and 14 days, the release of GDF5 (**A**), BB-1 (**E**), and BMP-2 (**I**) from ground cement discs was measured by ELISA and used for the calculation of cumulative release (**B**,**F**,**J**), % release (**C**,**G**,**K**), and retention (**D**,**H**,**L**). Data are expressed as means ± SEM (*n* = 3). * *p* ≤ 0.05 Mann–Whitney *U* test vs. pure CPC. + *p* ≤ 0.05 Mann–Whitney *U* test vs. 5% fibers.

**Figure 4 pharmaceutics-11-00455-f004:**
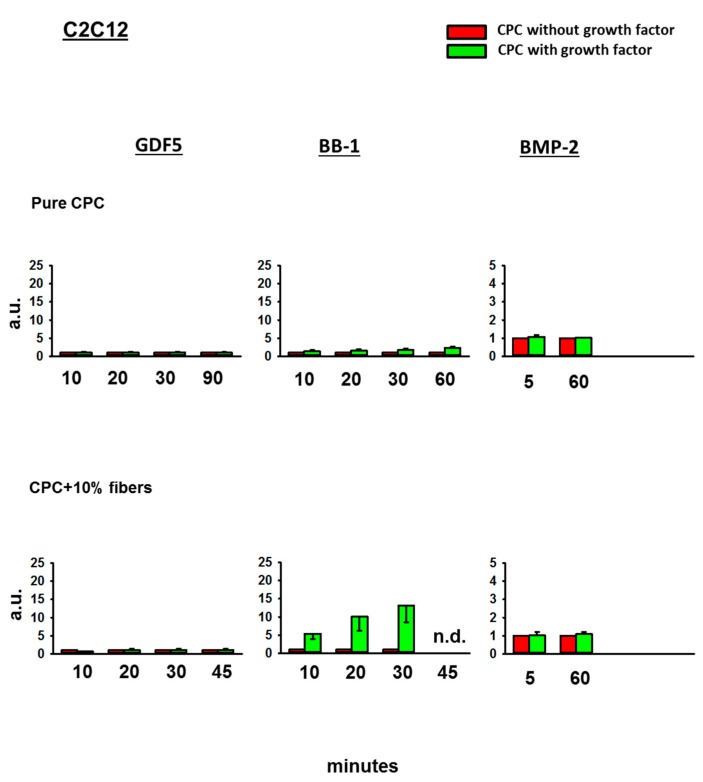
Alkaline phosphatase (ALP) activity in C2C12 cells. C2C12 cells were cultivated for 3 days in diluted extracts of GDF5, BB-1, or BMP-2-containing, ground pure CPC or PLGA fiber-reinforced CPC. Thereafter, the ALP activity was measured using an ALP assay and the data were expressed as fold-change induction compared to the CPC without growth factor; data are expressed as means ± SEM (*n* = 3); n.d. = not determined; a.u. = arbitrary units.

**Figure 5 pharmaceutics-11-00455-f005:**
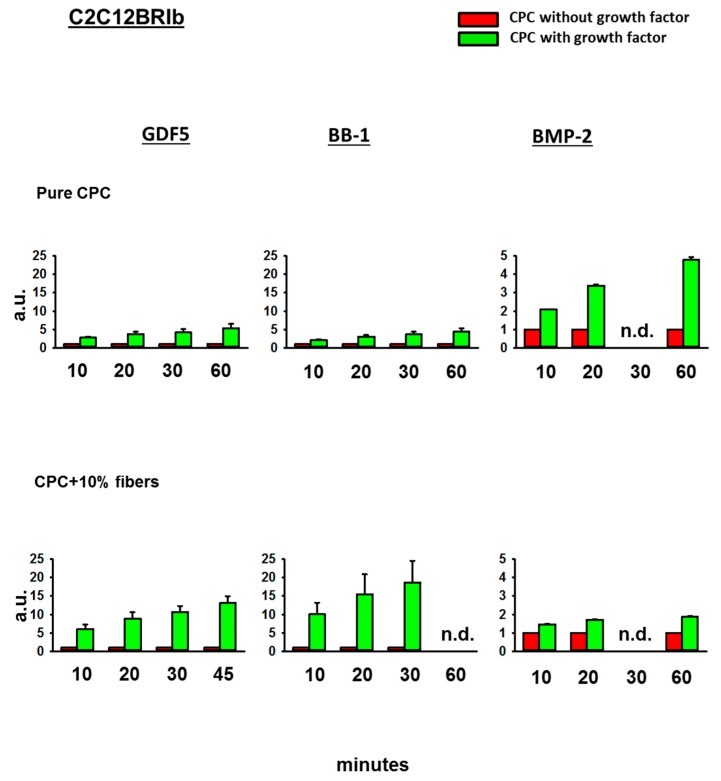
Alkaline phosphatase (ALP) activity in C2C12BRIb cells. C2C12BRIb cells were cultivated for 3 days in diluted extracts of GDF5, BB-1, or BMP-2-containing, ground pure CPC or PLGA fiber-reinforced CPC. Thereafter, the ALP activity was measured using an ALP assay and the data were expressed as fold-change induction compared to the CPC without growth factor; data are expressed as means ± SEM (*n* = 3); n.d. = not determined; a.u. = arbitrary units.

**Figure 6 pharmaceutics-11-00455-f006:**
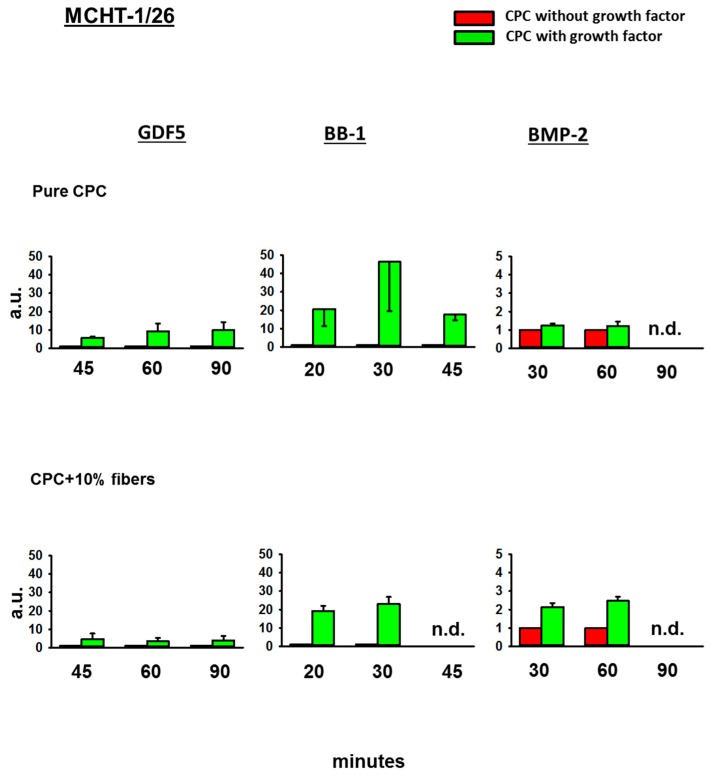
Alkaline phosphatase (ALP) activity in MCHT-1/26 cells. MCHT-1/26 cells were cultivated for 3 days in diluted extracts of GDF5, BB-1, or BMP-2-containing, ground pure CPC or PLGA fiber-reinforced CPC. Thereafter, the ALP activity was measured using an ALP assay and the data were expressed as fold-change induction compared to the CPC without growth factor; data are expressed as means ± SEM (*n* = 3); n.d. = not determined; a.u. = arbitrary units.

**Figure 7 pharmaceutics-11-00455-f007:**
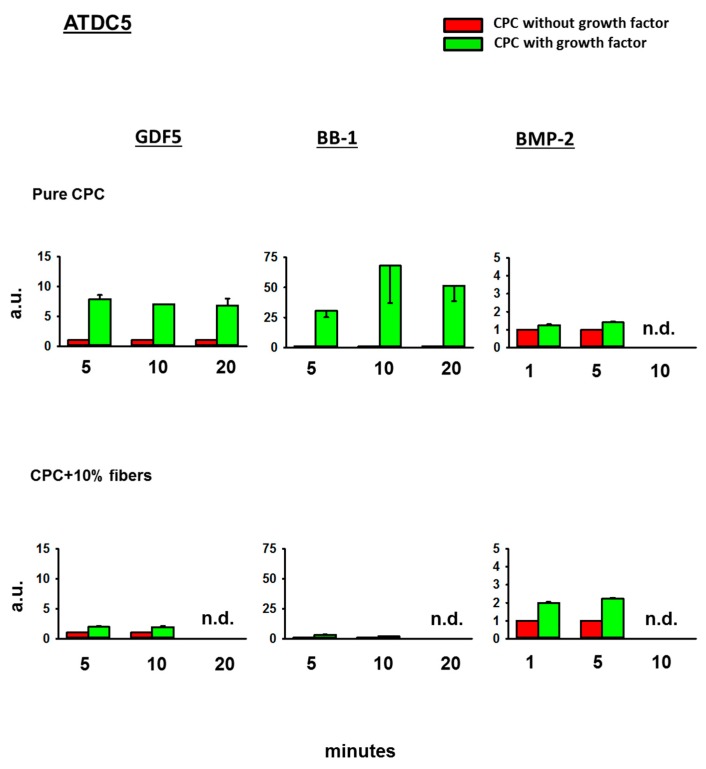
Alkaline phosphatase (ALP) activity in ATDC5 cells. ATDC5 cells were cultivated for 3 days in diluted extracts of GDF5, BB-1, or BMP-2-containing, ground pure CPC or PLGA fiber-reinforced CPC. Thereafter, the ALP activity was measured using an ALP assay and the data were expressed as fold-change induction compared to the CPC without growth factor; data are expressed as means ± SEM (*n* = 3); n.d. = not determined; a.u. = arbitrary units.

**Figure 8 pharmaceutics-11-00455-f008:**
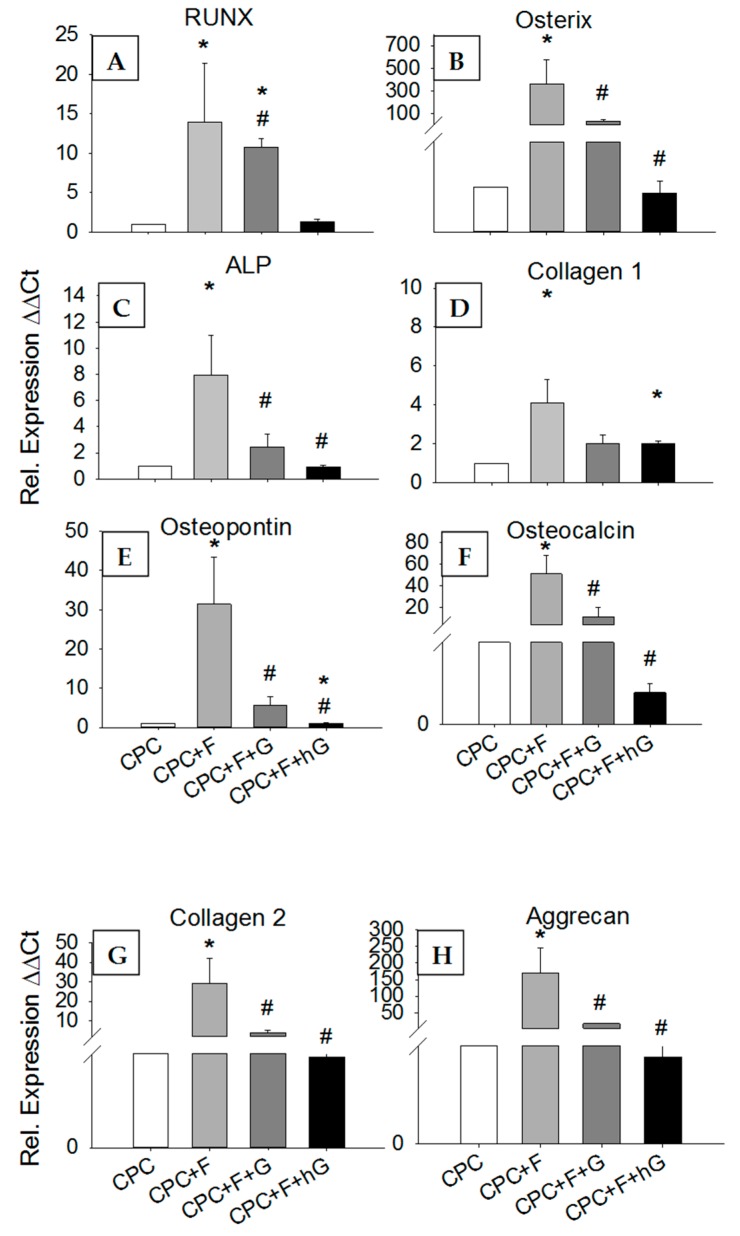
Effects of GDF5 extracts on the gene expression in hASCs. hASCs were seeded in 12-well plates and exposed for 3 days to the diluted extracts of CPC discs without fibers (CPC) and with 10% (*w*/*w*) fiber content (CPC+F), the latter either without GDF5 or doped with low dose (10 µg/mL; CPC+F+G) or high dose GDF5 (200 µg/mL; CPC+F+hG). Thereafter, the mRNA expression of Runx2 (RUNX; **A**), osterix (**B**), alkaline phosphatase (ALP; **C**), type I collagen (**D**), osteopontin (**E**), osteocalcin (**F**), type II collagen (**G**), and aggrecan (**H**) was measured using RT-PCR and the data were expressed as relative expression (as determined using the ΔΔCt method) compared to the CPC without fibers and growth factor; data are expressed as means ± SEM (*n* = 8); * *p* ≤ 0.05 Mann–Whitney U test vs. CPC; # *p* ≤ 0.05 Mann–Whitney U test vs. CPC+F.

**Figure 9 pharmaceutics-11-00455-f009:**
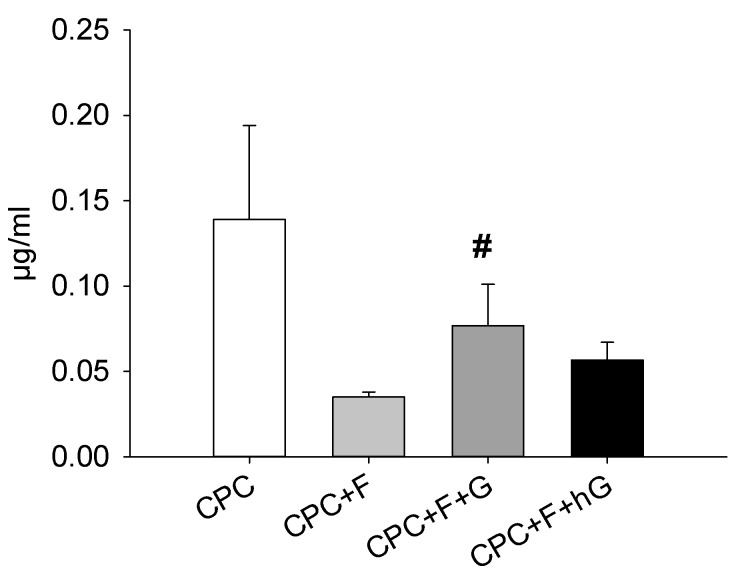
Effects of GDF5 extracts on the intracellular concentration of collagen 1 protein in hASCs. hASCs were seeded in 12-well plates and exposed for 3 days to the diluted extracts of CPC discs without fibers (CPC) and with 10% (*w*/*w*) fiber content (CPC+F), the latter either without GDF5 or doped with low dose (10 µg/mL; CPC+F+G) or high dose GDF5 (200 µg/mL; CPC+F+hG). Thereafter, collagen 1 concentrations in cell lysates were quantified using a commercial ELISA kit. The data are expressed as means ± SEM (*n* = 8); # *p* ≤ 0.05 Mann–Whitney U test vs. CPC+F.

**Table 1 pharmaceutics-11-00455-t001:** Primer sequences, annealing (*T*_annealing_) and melting (*T*_melting_) temperatures, and gene accession numbers applied in RT-PCR.

Gene	Forward/Reverse Primers	*T*annealing °C	*T*melting °C	Accession Number
Runx2 Runt-related transcription factor 2	FW: gccttcaaggtggtagccc Rev: cgttacccgccatgacagta	60	79	NM_001024630
Osterix	FW: tgcttgaggaggaagttcac Rev: aggtcactgcccacagagta	60	78	NM_001173467
Alkaline Phosphatase	FW: gcacctgccttactaactc Rev: agacacccatcccatctc	60	79	NM_000478
Collagen I	FW: tggagcaagaggcgagag Rev: caccagcatcacccttagc	60	85	NM_000088
Osteopontin (OPN)	FW: ttgcagtgatttgcttttgc Rev: gaacacgcatctgggtattt	60	82	NM_001040058
Osteocalcin	FW: gcaaaggtgcagcctttgtg Rev: ggctcccagccattgatacag	60	79	NM_199173
Collagen II	FW: tagggccggtctgcttcttgtaaa Rev: acatcaggtcaggtcagccattca	60	78	NM_001844
Aggrecan	FW: tctgtcaggcaaatctgggatggt Rev: atgccacttggtaggccact	60	78	NM_001135
Aldolase	FW: tcatcctcttccatgagacagtct Rev: attctgctggcagatactggcataa	58	82	NM_000034

**Table 2 pharmaceutics-11-00455-t002:** Differences among the release of the different BMPs from ground pure CPC and CPC ± PLGA fibers (GDF5, BB-1, BMP-2; *n* = 3 experiments each); **#**
*p* ≤ 0.05 vs. GDF5; **§**
*p* ≤ 0.05 vs. BB-1; note: differences among the different release time points are only shown in Figure 3.

		GDF5					BB-1					BMP-2				
**Time point (days)**		**1 h**	**1**	**3**	**7**	**14**	**1 h**	**1**	**3**	**7**	**14**	**1 h**	**1**	**3**	**7**	**14**
**Release (%)**	Pure CPC	9.68	14.14	16.39	17.13	17.64	5.35^#^	9.19	10.54^#^	10.94^#^	11.29^#^	5.42^#^	13.84^#§^	21.09^#§^	24.36^#§^	25.71^#§^
	CPC + 5% fibers	11.57	16.45	17.88	18.24	18.54	5.30^#^	8.99	10.12	10.44	10.71	4.78^#^	15.94^#§^	25.73^#§^	31.63^#§^	33.92^#§^
	CPC + 10% fibers	12.93	19.99	21.88	22.32	22.59	6.21^#^	10.85^#^	12.58	12.96	13.18^#^	6.26^#^	26.06^#§^	36.23^#§^	41.34^#§^	43.72^#§^
		**GDF5**					**BB-1**					**BMP-2**				
**Time point (days)**		**1 h**	**1**	**3**	**7**	**14**	**1 h**	**1**	**3**	**7**	**14**	**1 h**	**1**	**3**	**7**	**14**
**Release (ng/mL)**	Pure CPC	968.21	445.87	225.50	73.90	50.84	535.58^#^	383.70	135.57^#^	39.17^#^	35.67^#^	542.95^#^	841.53^#§^	724.80^#§^	326.84^#§^	135.20^#§^
	CPC + 5% fibers	1157.43	488.07	142.64	36.64	29.30	530.47^#^	369.27	112.37	32.16	26.89	478.83^#^	1115.30^#§^	979.73^#§^	590.00^#§^	228.97^#§^
	CPC + 10% fibers	1293.16	705.91	189.64	43.64	26.90	621.76^#^	464.14^#^	172.40	38.41	21.42^#^	626.51^#^	1979.53^#§^	1017.47^#§^	511.34^#§^	237.37^#§^

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
