# Peer review of "In Vitro Release of Bioactive Bone Morphogenetic Proteins (GDF5, BB-1, and BMP-2) from a PLGA Fiber-Reinforced, Brushite-Forming Calcium Phosphate Cement"

_pharmaceutics, 2019, doi:10.3390/pharmaceutics11090455_

Round 1

Reviewer 1 Report

The authors have described in details the fabrication of discs and cuboids from brushite-forming calcium phosphate cement including PLGA fiber-reinforced, but in the Results section the visualization of these materials is absent. So I can recommend including microscopy images demonstrating the cement ultrastructure and interaction of cement material with PLGA fibers using scanning electronic microscopy or atomic force microscopy or other visualization methods. Also it is desirable to include some information about the mechanical properties of studied materials before and after loading with bioactive proteins. 

Moreover, the mechanism of cement loading with bioactive proteins remains unclear, so I recommend to include a scheme of this process to the article.

Authors have claimed that the bioactive materials described in this manuscript can be used for bone regeneration. So it will be informative to show the possibility of growth and distribution of osteogenic cell lines on studied materials using scanning electronic microscopy or other visualization methods.

Although the authors have provided the litrature review background, some recent important studies of cements, PLGA and another mineral-organic tissue engineering scaffolds need to be cited in presented manuscript: 

DOI: 10.1007/s10856-018-6114-9 

DOI: 10.1016/j.jphotobiol.2017.06.002

DOI: 10.1039/C6RA26020A

DOI: 10.1007/s10856-016-5665-x, 

DOI: 10.1039/C6NR00641H

Author Response

Point-to-point-response to the Reviewers

We thank the reviewers for their critical evaluation of our manuscript ‘In vitro release of bioactive bone morphogenetic proteins (GDF5, BB-1, and BMP-2) from a PLGA fiber-reinforced, brushite-forming calcium phosphate cement’ and the constructive comments. We hope that the point-to-point response will render the manuscript acceptable for publication in Pharmaceutics.

REFEREE REPORT(S):

Referee: 1

Comments and Suggestions for Authors

The authors have described in details the fabrication of discs and cuboids from brushite-forming calcium phosphate cement including PLGA fiber-reinforced, but in the Results section the visualization of these materials is absent. So I can recommend including microscopy images demonstrating the cement ultrastructure and interaction of cement material with PLGA fibers using scanning electronic microscopy or atomic force microscopy or other visualization methods.

- Our group has previously published such images. This is now specified in the Introduction (lines 70 -73 of the revised version), including the introduction of one new own reference (new ref. 27).

Also it is desirable to include some information about the mechanical properties of studied materials before and after loading with bioactive proteins.

- Unfortunately, such data are not available. However, extensive biomechanical characterization of the pure CPC is reported in reference [26] and the new reference [27].

Moreover, the mechanism of cement loading with bioactive proteins remains unclear, so I recommend to include a scheme of this process to the article.

- The process of protein loading is now described in more detail in the Methods (lines 134 -138 of the revision), hopefully clarifying the procedure.

Authors have claimed that the bioactive materials described in this manuscript can be used for bone regeneration. So it will be informative to show the possibility of growth and distribution of osteogenic cell lines on studied materials using scanning electronic microscopy or other visualization methods.

- Again, such images have been published by our group previously and are now referenced in the Introduction (lines 73 - 74), again including a recently accepted own reference (new ref. 8).

Although the authors have provided the litrature review background, some recent important studies of cements, PLGA and another mineral-organic tissue engineering scaffolds need to be cited in presented manuscript:

DOI: 10.1007/s10856-018-6114-9

DOI: 10.1016/j.jphotobiol.2017.06.002

DOI: 10.1039/C6RA26020A

DOI: 10.1007/s10856-016-5665-x,

DOI: 10.1039/C6NR00641H

- These references and a recent review (new refs. 1-6) are now included in an initial paragraph on new synthetic biomaterials for bone repair and replacement in the Introduction (lines 45 – 55).

Reviewer 2 Report

In this study, author reported in vitro release of BMP family protein from PLGA fiber reinforced CPC.  Several questions and commented are listed below:

Abstract did not show conclusion and significance of the result clearly.

The title of 2.3 and 2.4 is misled that BMP is studied. It should be BMP family proteins

Why author did not do BMP-2 release expt in CPC (without PLGA fiber) as GDF-5 and BB1 in 2.3 and 2.4?

How to calculate or measure % retention?

Author should clearly state the source and property of C2C12, MCHT1/26 and ATDC-5

ALP is not purely reflect osteogenic cell activity but it also express chondrogenic cell during mineralsation such as ATDC-5. Why did author mention that ATDC-5 is osteogenic? It is best that author use sheep bone marrow stem cell.

Apart from ALP activity, author should also do gene and protein expression of ostegenic marker protein such as Cbfa-1, Runx2, Collagen type alpha 1 and osteocalcin to reflect the bioactivity of released BMP family. And compared BMP family protein released from CPC as compared with pure growth factor applied on cells.  

PLGA will be degraded into glycolic and lactic acid in vivo. Does these acids affect the release of BMP family protein differently.

In the experiment, the sample size is only 3. It is not large enough to draw solid result and conclusion.

Author Response

Referee: 2

In this study, author reported in vitro release of BMP family protein from PLGA fiber reinforced CPC.  Several questions and commented are listed below:

Abstract did not show conclusion and significance of the result clearly.

- The arguments concerning the conclusion and significance of the results have now been expanded (lines 39 - 40).

The title of 2.3 and 2.4 is misled that BMP is studied. It should be BMP family proteins

- The headings of sections 2.3. and 2.4. have been modified (lines 132; 149).

Why author did not do BMP-2 release expt in CPC (without PLGA fiber) as GDF-5 and BB1 in 2.3 and 2.4?

- The first release studies in sections 2.3. and 2.4. were focused on the osteoinductive molecule GDF5 and its potent mutant protein BB-1, which are currently under investigation in our group for potential clinical development. In section 2.5., the results for these 2 molecules were then compared to those of the already clinically applied BMP-2.

How to calculate or measure % retention?

- The calculation of the % retention is now explained in more detail (lines 167 - 170).

Author should clearly state the source and property of C2C12, MCHT1/26 and ATDC-5

- The source of the cell lines is now clearly stated in the respective paragraph (lines 176 - 179).

ALP is not purely reflect osteogenic cell activity but it also express chondrogenic cell during mineralsation such as ATDC-5. Why did author mention that ATDC-5 is osteogenic? It is best that author use sheep bone marrow stem cell.

- The term ‘osteogenic activity’ in section 2.6. was replaced by ‘biological activity’ in order to address the broad differentiation potential of the cell lines (line 182).

Investigation of sheep bone marrow stem cells is unfortunately beyond the scope of the present manuscript. However, the biological activity of the CPC extracts has now been analyzed in human adipose tissue-derived mesenchymal stem cells (hASCs; see below).

Apart from ALP activity, author should also do gene and protein expression of ostegenic marker protein such as Cbfa-1, Runx2, Collagen type alpha 1 and osteocalcin to reflect the bioactivity of released BMP family.

- The bioactivity of the BMP extracts has now been confirmed by the analysis of their effects on the gene and protein expression of osteogenic and chondrogenic markers in hASCs (lines 28 – 30 and 36 – 37; Methods, sections 2.7. to 2.9.; lines 405 – 438; new Figs. 8 and 9; lines 531 - 536). This has led to the inclusion of an additional author from our group (Victoria Horbert).

And compared BMP family protein released from CPC as compared with pure growth factor applied on cells.

- All ALP bioactivity assays included a standard dilution curve of the respective free, non-carrier bound BMPs (data not shown). On the basis of these data, an exemplary estimate of the concentrations of bioactive BMP released in 3 day extracts is now attempted for GDF5 (lines 550 – 563).

PLGA will be degraded into glycolic and lactic acid in vivo. Does these acids affect the release of BMP family protein differently.

- To our knowledge, there are no published data on differential effects of glycolic versus lactic acid on the release of the different BMPs analyzed in the present study. The PLGA fibers applied in the present study were chosen on the basis of their established clinical use as suture material and a potential clinical development. This point is now briefly touched in the Methods (section 2.1.) and the Discussion (lines 461 – 464).

In the experiment, the sample size is only 3. It is not large enough to draw solid result and conclusion.

- As mentioned above, the bioactivity of the BMP extracts has now been confirmed by the analysis of their effects on the gene and protein expression of osteogenic and chondrogenic markers in hASCs.

Round 2

Reviewer 1 Report

The revised MS can be published